# Anti-microRNA-1976 as a Novel Approach to Enhance Chemosensitivity in *XAF1*^+^ Pancreatic and Liver Cancer

**DOI:** 10.3390/biomedicines11041136

**Published:** 2023-04-10

**Authors:** Tsai-Yen Lee, Chien-Jen Tseng, Jin-Wun Wang, Ching-Po Wu, Chin-Yuan Chung, Ting-Ting Tseng, Shao-Chen Lee

**Affiliations:** 1School of Medicine, Fu Jen Catholic University, New Taipei City 24205, Taiwan; 2Department of Gastroenterology and General Surgery, ChiMei Hospital, Tainan City 72263, Taiwan; 3Department of Surgery, ChiMei Hospital, Tainan City 72263, Taiwan

**Keywords:** miR-1976, exosome, chemoagent, hepatoma, pancreatic cancer, XAF1

## Abstract

The current cancer treatments using chemoagents are not satisfactory in terms of outcomes and prognosis. Chemoagent treatments result in cell death or arrest, but the accompanying cellular responses are not well-studied. Exosomes, which are extracellular vesicles secreted by living cells, might mediate cellular responses through microRNAs. We found that miR-1976 was highly enriched in exosomes secreted after chemoagent treatment. We developed a novel approach for in situ mRNA target screening and discovered several miR-1976-specific mRNA targets, including the proapoptotic gene XAF1, which was targeted by miR-1976 and which suppressed chemoagent-induced cell apoptosis. Increased RPS6KA1 gene transcription was associated with the increase in its intronic pre-miR-1976 expression. Blockade of miR-1976 could enhance chemosensitivities of hepatoma and pancreatic cancer cells in an XAF1-dependent manner, as evidenced by increased levels of cell apoptosis, reduced IC50 in cell toxicity assays, and suppressed tumor growth in animal xenograft experiments in vivo. We propose that intracellular levels of miR-1976 determine chemosensitivity, and its blockade could be a novel strategy and potential therapeutic application in cancer treatment.

## 1. Introduction

MicroRNAs (miRNAs) are ribonucleotides with a length of 20–24 nts that post-transcriptionally regulate protein expression and function [1,2,3], and this is correlated with many physiological or pathological processes [4,5]. Exosomes are exocytotic vesicles that are presented in extracellular fluid, plasma, or urine samples and are taken up by defined cell targets [6,7]. The presence of specific proteins, lipids, mRNAs [6], or miRNAs [8] in exosomes make them novel regulators that are involved in several physiological or pathological processes [9,10,11]. Recent advances in the studies of circulating miRNAs also suggest that exosomal miRNAs would act as potential regulators [12] or pathological markers clinically [13,14,15].

Chemoagents induce tumor cell apoptosis and suppress tumor growth. It has been suggested anti-cancer drugs cause the release of exosomes with heat shock proteins from human hepatocellular carcinoma cells [16]. They elicit effective natural killer cell antitumor responses that explain the failure of immune surveillance, high incidence of recurrence, and poor prognosis of hepatocellular carcinoma in vitro. However, whether other defined mediators released from tumor cells, such as miRNAs, would affect the outcomes of chemotherapy is not clear.

The *XAF1* gene product is a novel inhibitor protein, called XIAP-associated factor 1 (XAF1), which has been shown to antagonize the anticaspase activities of XIAP [17]. XAF1 is transcriptionally regulated by p53. However, it stabilizes p53 through competition with MDM2 [18]; thus this feedback loop would control cell apoptosis and inhibit p53–p21-directed cell cycle arrest [18]. Thus, XAF1 has been recognized as a tumor suppressor, and its expression is generally downregulated in several cancer cells [19].

In this paper, we show the chemoagent-perturbed expression of miRNAs and that its regulation contributes toward achieving chemoresistance. This mechanism contributes to a strategy for potential therapeutic application.

## 2. Materials and Methods

### 2.1. Materials

The AT-1976 is the miR-1976-specific antagomir with characteristics of 2′-*O*-methylation, modified phosphorothioate linkage, and 3′-cholesterol addition [20] and was purchased from MDBio, Inc, Taiwan. The following sequence was used: 5′- c_s_c_s_uccugcccuccuugc_s_u_s_g_s_u_s_-Chol-3′. A control RNA with a sequence that does not bind to any known microRNAs was: 5′-a_s_u_s_gacuaucgcuauucgc_s_a_s_u_s_g_s_-Chol-3′. The subscript ‘s’ represents phosphonothioate linkage; ‘Chol’ represents a cholesterol group linked through a hydroxyprolinol linkage.

### 2.2. Cell Cultures, Transfection, and Treatments

Human hepatoma HepG2 and Hep3B cells and human pancreatic AsPC-1 and BxPC-3 cells were all purchased from the Bioresource Collection and Research Center, Taiwan. They were all maintained in culture dishes (Corning Incorporated Life Sciences, Glendale, AZ, USA) containing MEM or RPMI-1640 medium supplemented with 10% (*v*/*v*) fetal bovine serum (Biological Industries, Ltd., Kibbutz Beit-Haemek, Israel), 100 units/mL penicillin, and 100 μg/mL streptomycin under 5% CO_2_ with 100% humidity.

For miR-1976 overexpression, cells were transfected with miR-1976 overexpression plasmids using the TransIT-X2^®^ Dynamic Delivery System (Mirus Bio LLC., Medison, WI, USA) according to the manufacturer’s instructions. To examine exosome release, cell apoptosis analysis (Western blot), cell viability, and gene expression analysis, cells were treated with chemoagents for 48 h. For cell apoptosis analysis (annexin V–PI staining) and cell cycle analysis, cells were treated with treated with chemoagents for 24 h.

### 2.3. Exosome Preparation and Characterization

The exosome-producing medium was prepared to remove residual exosomes from FBS as referenced in [21], with modification. Briefly, 50% (*v*/*v*) FBS in DMEM or MEM medium was centrifuged at 2000× *g* for 10 min, and then centrifuged at 100,000× *g* (Beckman Optima L90-K with 90Ti rotor; Beckman Coulter Taiwan Inc., Taipei, Taiwan) for 16 h at 4 °C. The supernatant was collected and diluted to 10% (*v*/*v*) FBS with serum-free DMEM or MEM and sterile-filtered through a 0.22 µm filter.

For production of cell-derived exosomes, 2 × 10^6^ cells were seeded in culture medium overnight, which was then replaced with exosome-producing medium with or without 10 μM chemoagents and cultured for another 2 days. In total, 50 mL of exosome-containing medium was collected. The exosomes were isolated using a Total Exosome Isolation kit (Life Technologies-Thermo Fisher Scientific, Inc., Grand Island, NY, USA) according to the suggested protocol. The pellets containing secreted exosomes were further washed by diethylpyrocarbonate-treated PBS and centrifuged at 100,000× *g* for 60 min (Beckman Optima MAX-E with TLA-120.2 rotor; Beckman Coulter Taiwan Inc., Taipei, Taiwan); these steps were repeated twice to completely remove residual serum protein. The protein content in the exosome solution was determined using protein assay reagents from Thermo Fisher Scientific Inc. (Pittsburgh, PA, USA).

Exosomes were characterized for their sizes by transmission electron microscopy analysis. The preparation of grids containing cell-derived exosomes was performed accordingly [21]. In general, 5 µL of paraformaldehyde-fixed exosome solution was coated onto a formvar–carbon-coated electron microscopy grid (Electron Microscopy Sciences, Hatfield, PA, USA). The contrast staining by uranyl oxalate solution and successive methyl cellulose–uranyl acetate solution was conducted on ice. The imaging of exosomes was performed by JEOL JEM-1400 transmission electron microscopy (JEOL USA Inc., Peabody, MA, USA) at 80 kV.

### 2.4. In Situ mRNA Target Screening by miR-1976-Specific Cloning

In order to identify the specific mRNA targets regulated by miR-1976, we established a novel approach to amplify potential mRNA targets in situ by miR-1976-specific cloning. Basically, one miR-1976-mimicked sequence (5′-cctcctgccctccttgctgt-3′) was used as a primer in reverse transcription to build up miRNA-1976-targeted cDNA pools. Total RNA (1 ug) from Hep3B or HepG2 cells and the miR-1976 primer (2.5 μM) were used in reverse transcription reactions with the MMLV HP reverse transcriptase (Epicentre, Madison, WI, USA). The cDNAs were further amplified by Taq polymerase (Ampliqon, Skovlunde, Denmark) in 30 repeated cycles (94 °C, 30 s; 55 °C, 30 s; 72 °C, 30 s;). Terminal modification was then carried out with 1 unit of Taq polymerase and 0.2 mM of dATP. These DNA segments were then ligated into the pTA-TOPO vector using the TA TOPO Cloning Kit (Thermo Fisher Scientific Inc., Pittsburgh, PA, USA). The plasmids were then transformed into DH5a bacteria and cultured on X-gal/IPTG-containing agar plates. After overnight culture, white colonies were picked and further cultured in 96-well-plates for further validation by PCR.

To confirm the presence of the inserted segment in individual clones, colony PCR was performed using the forward primer: 5′-gtaaaacgacggccag-3′ and reverse primer: 5′-caggaaacagctatgac-3′. After isolation of the plasmid from individual bacterial clones, an additional PCR was carried out to confirm the segment insertion before DNA sequencing. The DNA sequence of each clone was then compared with known gene sequences by BLAST analysis using Blastn in https://blast.ncbi.nlm.nih.gov (accessed on 29 January 2023). The database was set up for the human genome plus transcripts. For each potential miR-1976 target, the binding duplex (mRNA–miRNA) and folding energy were presented using RNA22 [22]. RNA22 adapts a pattern-based algorithm to determine miRNA-targeting sites on a user-defined nucleotide sequence without cross-organism conservation constraints [22]. The sequenced clones, mRNA–miRNA duplexes, and folding energies derived from Hep3B or HepG2 cells are listed in Appendix A.

### 2.5. Plasmid Construction and Preparation

The recombinant DNA experiments were performed according to the National Institutes of Health Guidelines and were reviewed by the Fu Jen Catholic University Biosafety committee. For constructing the miR-1976 overexpression plasmid, the primers used to construct the pri-miR-1976 sequence were 5′-cctgatagatctgcagcaaggaaggcaggggtcctaaggt gtgtcctcct-3′ and 5′-cctgatgg atccacagcaaggagggcaggaggacacaccttaggacccct-3′. The PCR product was then digested by *Bgl*II and *Hind*III (Thermo Fisher Scientific Inc., Pittsburgh, PA, USA) and ligated with the pcDNA6.2-EmGFP plasmid (Thermo Fisher Scientific Inc., Pittsburgh, PA, USA) digested with the same enzymes.

Cloning of the *XAF1_*3’UTR sequence was performed by PCR using the Q5_hotstart enzyme mixture, cDNA pools from HepG2 cells, and specific primers: 5′-cctagtggtaccatttggaaaaggaaaggtactaca aattcaaaagatttc-3′ and 5′-cctgatacta gtgtagagatctttcatctccctgg-3′. The PCR product was digested by *Hind*III and *Spe*I, and then ligated into the pMIR-REPORT™ luciferase vector (Thermo Fisher Scientific Inc., Pittsburgh, PA, USA) predigested with the same enzymes. The ligated plasmids were then transformed into HIT^TM^-DH5α cells (Real Biotech Corporation, New Taipei City, Taiwan) and single bacterial colonies were selected and amplified. The plasmids were isolated from the bacteria and sequence-characterized to confirm the fidelity of clones.

### 2.6. In Vitro Luciferase Reporter Assay

Luciferase activity assays were performed using the pMIR-REPORT™ miRNA expression reporter vector system (Thermo Fisher Scientific Inc., Pittsburgh, PA, USA). The vectors containing the luciferase gene with or without the *XAF1*_3′UTR were transfected into cells and the luciferase activities were compared in the presence of miRNA-1976 expression. In brief, Hek293 cells were seeded (4 × 10^4^ cells per well) into 24-well plates till 70~80% confluence. A total of 1 µg DNA containing the luciferase–UTR construct, β-Gal vector, and miRNA-expression vector in the ratio of 1:1:10 were transfected using the Turbofect transfection reagent (Thermo Fisher Scientific Inc., Pittsburgh, PA, USA). Cells were then harvested using trypsin/EDTA, and the β-Gal activity and luciferase activity were assayed using a luciferase assay system (Promega corporation, Madison, WI, USA). Experiments were repeated four times and luciferase activities were normalized against β-Gal activity.

### 2.7. Western Blot and Antibodies

For Western blotting analysis, the cells (60–80% confluency) were washed twice by PBS and scraped. The cells (1 × 10^6^ cells) were disrupted by the lysis buffer (10 mM Tris-HCl, 5 mM EDTA, pH 8.0, phosphatase inhibitors, and protease inhibitors) and kept on ice for 30 min. The protein mixture was subjected to SDS-PAGE and transferred onto a PVDF membrane followed by blocking with 5% (*w*/*v*) skim milk. The membrane was then incubated with primary antibodies (1:1000 in 5% skim milk in TBST) for 2 h at room temperature and horseradish peroxidase-conjugated secondary antibody (1:10,000) for 1 h at room temperature followed by enhanced chemiluminescent (EMD Millipore Co., Billerica, MA, USA) detection. The primary antibody against cleaved PARP (~89 kDa) was purchased from Cell Signaling Inc., Danvers, MA, USA. The primary antibodies against beta-actin (~42 kDa), TP53 (~53 kDa), XAF1 (~38 kDa), and MDM2 (~90 kDa) were purchased from GeneTex Inc., Hsinchu City, Taiwan. The experiments were repeated at least twice, and representative figures are shown.

### 2.8. RNA Extraction and microRNA Microarray Analysis

The miRNAs were isolated from 400 μg of exosomes enriched from the conditioned medium of untreated or CPT-treated hepatoma HepG2 cells using an miRNA isolating kit (Geneaid Biotech Ltd., New Taipei City, Taiwan). The small RNAs were characterized by the Agilent RNA 6000 Nano assay kit (Agilent Technologies, Santa Clara, CA, USA) to ensure good quality. The small RNAs were set up for probe hybridization on a microarray chip (HmiOA5.1; Phalanx Biotech Group, Hsinchu City, Taiwan) according to the Phalanx miRNA hybridization protocol. Two repeated hybridizations were performed, and the intensity values from repeated probes within one chip were combined to obtain the median. Coefficient of variance for miRNA from repeated probes within one chip were calculated and normalized. The log_2_ ratio values for miRNAs in CPT-treated group vs. untreated group were calculated to indicate the fold change. Differentially expressed miRNA candidates were selected using a threshold of fold change ≥ 0.8 or ≤−0.8 and a *p*-value ≤ 0.05. For advanced data analysis, intensity data were pooled and calculated to identify differentially expressed miRNAs based on the threshold of fold change and *p*-value. The correlation of expression profiles between samples and treatment conditions was demonstrated by unsupervised hierarchical clustering analysis. The array data was deposited in the NCBI Gene Expression Omnibus database (http://www.ncbi.nlm.nih.gov/geo/, accessed on 10 November 2015) with the accession number GSE74829.

### 2.9. qPCR Analysis for miRNA Expression

The level of miRNAs in exosomes was analyzed by qPCR as referenced in [23]. The miRNA pools were extracted using RNAzol reagent (Molecular Research Center Inc., Cincinnati, OH, USA). PolyA tails were added onto small RNA pools using polyA polymerase (Epicentre-Illumina, Madison, WI, USA) at 37 °C for 30 min. The cDNAs were synthesized with MMLV HP reverse transcriptase (Epicentre-Illumina, Madison, WI, USA), using specified primers [23] and pooled polyA-added RNA (0.5 μg) as template. The qPCR was performed using VeriQuest Fast SYBR green qPCR reagent (Affymetrix Inc., Santa Clara, CA, USA) in a StepOne Plus real-time PCR system (Applied Biosystems-Thermo Fisher Scientific Inc., Grand Island, NY, USA). The 2^−ΔΔCT^ method was used to determine the relative gene expression using U6 as control. The DNA segment corresponding to mature miRNA was used as the forward primer, and the specific reverse primer was used accordingly [23].

### 2.10. Cell Viability Assay by Alamar Blue

Cultured cells were collected by trypsinization and counted with a hemocytometer. A total of 2 × 10^3^ cells were plated into a 96-well culture plate. The next day, the cells were treated with different agents as designated. The cells were then incubated and allowed to proliferate for three days. The cell viability (reflected by both the cell number and viability) was determined by the Alamar Blue assay (AbD Serotec., Inc., Raleigh, NC, USA), which evaluates mitochondrial activity. The Alamar Blue reagent is used as an oxidation–reduction indicator that undergoes a colorimetric change, as well as a fluorescent change, in response to cellular metabolic reduction. The change in intensity is proportional to the quantity of living cells respiring.

### 2.11. Flow Cytometry

Cell apoptosis was analyzed by flow cytometry using the Annexin V-FITC Apoptosis Detection Kit (Dojindo laboratories, Kumamoto, Japan). Basically, 1 × 10^5^ cells were treated with chemoagents for 24 h, then the cells were trypsinized and collected for annexin V and PI labeling according to the manufacturer’s protocol. The fluorescent-labeled cells were then analyzed by Gallios Flow cytometry (Beckman Coulter Life Sciences, Indianapolis, IN, USA). Cell cycle analysis was performed.

### 2.12. Animal Xenograft Experiments

The protocol and design of the animal experiments were reviewed and approved by the Experimental Animal Care and Use Committee in Fu Jen Catholic University (approval number A10659). Basically, 1 × 10^6^ pancreatic BxPC-3 or AsPC-3 cells (in 100 uL of serum-free medium and matrigel) were subcutaneously inoculated into 5-week-old nude mice (BALB-c/Nu strain, female), and the mice were supplemented with sufficient food and water in a specific-pathogen-free environment. Each group contained 5 mice to obtain statistically significant results. Treatment comprised an i.p injection of the relevant drug combination three times per week using the dosage of 100 mg/kg GEM with or without AT-1976 cotreatment (1.6 mg/kg or 6.4 mg/kg). Tumor size was measured every week with Vernier calipers, and tumor volume and tumor inhibition rate (%) were calculated according to the following formula: V = 0.5 × longest diameter × shortest diameter^2^.

### 2.13. Statistical Analysis

Statistical analysis was performed with the Origin7.0 software (OriginLab Corporation, Northampton, MA, USA) using the paired/one-tailed two-sample *t*-test. A *p*-value of <0.05 or <0.01 was considered statistically significant and is indicated accordingly.

## 3. Results

### 3.1. miR-1976 Is Enriched in Exosomes Secreted from Damaged Hepatoma Cells

Damage of cancer cells by chemoagents is a general phenomenon upon chemotherapy; however, the components released from damaged cancer cells and the consequences they contribute to tumor cells are not clear. Exosomes are constantly or dynamically released from cells to maintain cell survival or to cope with their environment. They are also recognized as regulatory vesicles secreted from donor cells and can be accepted by original cells or other acceptor cells. Inside the exosomes, microRNA components have the potential to be post-transcriptional modifiers that regulate the protein expression and functions of acceptor cells.

Firstly, we wanted to explore whether specific components, such as microRNAs, are presented in damaged tumor cells. We isolated exosomes secreted from hepatoma HepG2 cells with or without treatment of the chemoagent camptothecin (CPT). Exosomes secreted from hepatoma cells were isolated and characterized by transmission electron microscopy and Western blotting [24]. Positive–negative contrast staining showed that these exosomes were approximately 30 nm in diameter, while the size of exosomes typically range from 30 to 100 nm, depending on the cell source. Western blotting also showed the presence of the exosome-specific marker CD63 [25] in these isolated exosomes.

Upon CPT treatment, elevated levels of cleaved PARP were observed, indicating severe cell apoptosis. The microRNAs in the exosomes were extracted and then examined by microarray analysis. The relative contents of exosomal miRNAs released from cells with or without chemoagent treatment were compared. Of all 1308 miRNAs examined, three miRNAs (hsa-miR-1976, -4728-3p, and -877-3p) were upregulated with a fold change in excess of 0.8, and six miRNAs (hsa-miR-4497, -204-3p, -6126, -5787, -1273e, and -1908-3p) were downregulated in the exosome released from CPT-treated hepatoma cells (Figure 1A). Clustering was performed to visualize the correlations among the replicates and varying sample conditions. Up- and down-regulated miRNAs were represented in consistent correlation, respectively. The highly enriched miRNAs in CPT-treated hepatoma-derived exosomes might play roles in surrounding hepatoma cells or distant cells at other origins through secreted exosomes. We hypothesized that these miRNAs might regulate gene expression and act cooperatively at acceptor cells.

Of all the upregulated miRNAs in CPT-treated hepatoma-derived exosomes, miR-1976 is the most significant one. In order to identify the specific mRNA targets regulated by miR-1976, we established a novel approach to amplify potential mRNA targets in situ by gene-specific cloning.

Basically, one miR-1976-mimicked sequence was used as a primer in reverse transcription to build up miRNA-1976-targeted cDNA pools. The logic of this gene-specific cloning is based on the algorithm of miRNA–mRNA duplex formation. Once this miR-1976-mimicked DNA binds to its specific target mRNA, reverse transcriptase can extend and generate the miR-1976-specific cDNA pool. The miR-1976 binding location will also be revealed, and the success or effectiveness of cloning is strongly dependent on the actual expression levels of target mRNAs, which reflects the abundance of miRNA-targeting mRNA in situ. The cDNA clones were prepared from cDNA pools by TA-cloning. The miR-1976-specific target clones were sequenced, and the target sequences were analyzed and identified by BLAST analysis.

The isolation, validation, and sequence analysis of miR-1976-targeted segments were performed (Figure 1B). DNA segments of miRNA-1976-specific genes originating at miR-1976 binding sites were generated by PCR using random primers and Taq DNA polymerase. These segments were then ligated into TOPO-TA cloning vectors, and plasmids with the miR-1976-targeted clones were isolated. A total of 48 clones were generated using the miR-1976-specific cDNA pool from Hep3B cells, while 68 clones were generated using the miR-1976-specific cDNA pool from HepG2 cells. A total of 18 plasmids generated from Hep3B cells and 24 plasmids generated from HepG2 cells were isolated and sequenced using the M13 reverse primer. The gene identities were determined using nucleotide BLAST. Of them, seven genes (*ZNF644*, *LENG8*, *GPATCH4*, *IPO8*, *ANKRD17*, *KTN1*, and *TPP1*) were identified from Hep3B cell-derived clones, while six genes (*TPP1*, *SLC20A2*, *RPL27*, *XAF1*, *WDR5*, and *SEC62*) were identified from HepG2 cell-derived clones. The other clones could not be identified as definite genes due to the failure of DNA sequencing, as these DNA fragments in human chromosomes were not validated as known genes. These 12 genes were experimentally identified by in situ mRNA target screening.

A further comparison was made to select potential miR-1976 targets associated with cell death. As seen in Figure 1C, the gene set associated with apoptosis-related cell death (1001 genes) and in silico-predicted genes regulated by miR-1976 (6201 genes) intersected with 12 gene targets determined experimentally. The *XAF1* and *SEC62* genes were mined by both in situ mRNA target screening and in silico target prediction using TargetScan [26,27]. However, only *XAF1* is associated with cell death through cell apoptosis. XAF1 is the inhibitor protein against XIAP, which suppresses cell apoptosis. That is, expression of the XAF1 protein may promote cell apoptosis. If *XAF1* is regulated by miR-1976, this could affect the levels of cell death. As seen in Figure 1D and Appendix A, the RNA duplex of the *XAF1* mRNA and miR-1976 showed partial complementary pairing. The binding site of miR-1976 at *XAF1* mRNA was predicted in silico and experimentally validated by clone sequencing.

Of note, this miR-1976 is classified as a mirtron [28,29,30], a microRNA originating from an intronic segment of an expressed gene. Pre-miR-1976 is generated from *RPS6KA1′*s intronic segment located between exon 7 and exon 8 of the *RPS6KA1* gene (Figure 1E). Transcription and debranching of the *RPS6KA1* mRNA would generate a hairpin product with a 5′ protrusion, known as a 5′-tailed mirtron [30].

### 3.2. miR-1976 Inhibits XAF1 Protein Expression and Suppresses Chemoagent-Induced Cell Apoptosis

Cell apoptosis is mainly determined by p53-mediated pathways. It has been reported that XAF1 is transcriptionally regulated by p53, and it competes for the MDM2-binding site in the p53 protein, and reversibly stabilizes the p53 protein. As seen in Figure 2A, XAF1 was expressed in the hepatoma HepG2 cells and pancreatic BxPC-3 cells but not in the hepatoma Hep3B cells and pancreatic AsPC-1 cells. This was dependent on the presence of p53 protein expression but not on that of the MDM2 protein.

Next, we constructed the miR-1976 expression plasmid and examined whether miR-1976 regulated XAF1 protein expression. As seen in Figure 2B, transfection of the miR-1976 expression plasmid increased miR-1976 expression 2-fold as examined by qPCR. The miR-1976 overexpression reduced XAF1 protein expression as investigated by Western blotting. In addition, the luciferase reporter assay (Figure 2C) showed a reduction in luciferase activity by the direct binding of miR-1976 to the predicted site at the 3′UTR of *XAF1* (located between 1532 and 1550). This suggested that miR-1976 directly targeted *XAF1*-3′UTR and suppressed XAF1 protein expression.

The levels of chemoagent-induced cell apoptosis were affected by the presence of miR-1976. As seen in Figure 2D, the amount of cleaved PARP (late apoptotic marker) in hepatoma HepG2 cells increased with treatment using increased CPT concentrations for 48 h. However, elevated levels of miR-1976 in HepG2 cells reduced the amount of cleaved PARP. In pancreatic BxPC-3 cells, gemcitabine (GEM) treatment also induced cell apoptosis, with higher levels of cleaved PARP in cells treated with lower GEM concentrations for 48 h. Indeed, elevated levels of miR-1976 in BxPC-3 cells reduced the amount of cleaved PARP. This implied that the presence of miR-1976 would reduce the sensitivity of chemoagent-induced cell apoptosis for hepatoma or pancreatic cells.

### 3.3. Decrease in Intracellular Levels of miR-1976 and Its Redistribution into Extracellular Vesicles

Next, we examined the miR-1976 levels inside the cells and in the exosomes with or without chemoagent treatment. As seen in Figure 3, the relative levels of miR-1976 inside the cells decreased upon chemoagent treatments, as seen in 4 different cells; meanwhile, those in the secreted vesicles were increased by the chemoagent treatments.

A decrease in specific miRNA levels might directly or indirectly regulate protein expression or mRNA levels through a block in translation and altered mRNA stability [31,32]. Since the intracellular amounts of miR-1976 decreased upon cell damage, we analyzed several survival-related gene expressions associated with *XAF1* (Figure 4). For all the hepatoma and pancreatic cancer cells tested, *XAF1* mRNA levels increased upon cell damage, which indicated a potency of cell apoptosis that was reasonable for chemoagent treatments. Levels of *TP53* mRNA were also increased by the chemoagent treatments as typical responses to trigger DNA repair or cell death upon chemoagent treatments. Although hepatoma Hep3B and pancreatic AsPC-1 cells are p53-negative cancer cells (see Figure 2A), the basal level of *TP53* mRNA increased relatively in these two cells, which explains the increased transcriptional levels of *XAF1*, *CDKN1A*, and *MDM2*. The *CDKN1A* gene encodes the p21cip1 protein, which is a cyclin-dependent kinase inhibitor that plays a critical role in regulating cell cycle progression and preventing uncontrolled cell proliferation. Extensively increased *CDKN1A* mRNA levels were observed in HepG2, AsPC-1, and BxPC-3 cells. Increased p21 expression can have a complex and context-dependent impact on cell behavior and plays both protective and deleterious roles in different biological processes. Upon chemoagent treatment, increased *CDKN1A* mRNA levels indicate the activation of a protective mechanism of cell cycle arrest by the p21cip1 protein. The MDM2 protein is the negative regulator of p53, and it inhibits the transcriptional activation activity of the p53 protein. As seen in Figure 4, MDM2 levels increased with chemoagent treatments, except in Hep3B cells. Increased MDM2 levels indicate the suppression of p53 transcription activities through feedback inhibition. These results suggest that the cells have contrasting responses to chemoagent treatments for both the protective pathway (cell cycle arrest for repair mechanism) and the death pathway (XAF1-associated cell apoptosis). In addition, *RPS6KA1* mRNA levels were relatively increased upon cell damage. *RPS6KA1* gene encodes one of the subunits of ribosomal S6 kinase, p90RSK (also called RSK1). p90RSK plays a role in the Ras–mitogen-activated protein kinase (MAPK) signaling pathway [33,34]. It is the downstream effector of Ras–extracellular signal-regulated kinase (ERK1/2) signaling. ERK1/2 activation directly phosphorylates and activates p90RSK and then activates different substrates. This promotes cancer cell proliferation/growth and suppresses cell apoptosis. Accompanied transcription and expression of *RPS6KA1* mRNA would also increase the level of the miR-1976 precursor.

### 3.4. Anti-miR-1976 Enhances Chemosensitivity in an XAF1-Dependent Manner

Since miR-1976 inhibits proapoptotic XAF1 protein expression and suppresses chemoagent-induced apoptosis, we investigated whether the anti-miR-1976 approach would enhance chemosensitivity. We designed an anti-miR-1976 molecule (AT-1976) using an antagomir RNA structure with the miR-1976 complementary sequence. Antagomir is an antisense oligonucleotide with a complementary sequence to a specific miRNA target. Its structure has modifications of 2’-methoxy groups at each ribose sugar, phosphorothioate bonds at the nucleotide backbone, and cholesterol conjugation at the 3′ end [35,36]. These modifications enable extended stabilities for in vivo application.

We treated hepatoma or pancreatic cancer cells with AT-1976 to determine whether cell death was promoted. As seen in Figure 5A, CPT induced cell apoptosis at concentrations of 1 and 10 μM, and the treatment of AT-1976 increased the content of cleaved PARP in hepatoma HepG2 cells. However, there was no increase in cleaved PARP content in hepatoma Hep3B cells treated with AT-1976 (Figure 5B). We hypothesized that this anti-miR-1976 strategy only applied to *XAF1*-positive cancer cells, since miR-1976 regulated XAF1 protein expression. As we examined cell apoptosis in pancreatic cancer cells with or without AT-1976 treatment, we observed the same *XAF1*-dependent effect. As seen in Figure 5C, GEM treatment significantly increased cell apoptosis at low GEM concentrations. Co-treatment with AT-1976 elevated cleaved PARP content in pancreatic BxPC-3 cells (*XAF1*-positive). For pancreatic AsPC-1 cells (*XAF1*-negative), GEM treatment did not cause a significant increase in cell apoptosis, even following co-treatment with AT-1976 (Figure 5D).

We also evaluated cell apoptosis by examining the outcome of annexin V–PI staining. As seen in Appendix A and Figure 5E, treatment with increased concentrations of chemoagents gradually increased the percentages of cell populations in section b (early apoptosis; Annexin V^+^/PI^−^), and section c (late apoptosis; Annexin V^+^/PI^+^). The addition of AT-1976 further increased the percentage of cell apoptosis. This strengthens our conclusion that blockade of miR-1976 would enhance chemoagent-induced cell apoptosis.

The effects of AT-1976 on chemoagent-induced cell toxicity were examined. For *XAF1*-positive HepG2 cells, AT-1976 co-treatment reduced the IC50 for CPT-induced cell toxicity from 0.12 μM to 0.026 μM (Figure 6A), while there was no significant difference in IC50 for *XAF1*-negative Hep3B cells with or without the presence of AT-1976 (Figure 6B). For *XAF1*-positive BxPC-3 cells, AT-1976 co-treatment reduced the IC50 for GEM-induced cell toxicity from 1.0 μM to 0.12 μM (Figure 6C); no significant difference in IC50 was observed for *XAF1*-negative AsPC-1 cells with or without the presence of AT-1976 (Figure 6D). These results suggested that AT-1976 antagonized miR-1976 and rescued the inhibition of XAF1 protein, thus promoting cell apoptosis and increased cell toxicities.

We also examine the effect of AT-1976 on cell cycle progression. As seen in Appendix A, GEM treatment reduced the G2 content in both BxPC-3 and AsPC-1 cells, which is consistent with the mechanism of GEM. In addition, AT-1976 co-treatment did not alter the profile of the cell cycle population with or without the presence of chemoagent treatment. This suggests that the mechanism of AT-1976 solely enhances cell apoptosis and death but does not block cell cycle progression.

Next, we investigated whether the application of AT-1976 had the potential to suppress tumor growth in vivo. Xenograft tumor models were established using either pancreatic BxPC-3 or AsPC-1 cells. Administration of GEM (at a dosage of 100 mg/kg) with or without AT-1976 was compared to the efficacy of AT-1976 co-treatment. As seen in Figure 7A, GEM suppressed tumor growth by 43% upon treatment for two weeks. Co-treatment of AT-1976 further reduced the tumor volume in a dose-dependent manner. An approximately two-fold efficacy (from 43% to 81% reduction) was achieved with the co-treatment of GEM and AT-1976 (at a dosage of 6.4mg/kg). However, there was no significant effect of AT-1976 in promoting GEM-mediated tumor suppression in AsPC-1 xenografts (Figure 7B). These results also support our aforementioned conclusion that such an anti-miR-1976 approach would enhance chemosensitivity in *XAF1*-positive tumors, at least in hepatoma and pancreatic cancer cells.

## 4. Discussion

Tumor cell injury by treatment with chemoagents is one of the major issues in chemotherapy [37]. The evaluation and dose modification of chemoagents, as well as the understanding of possible mechanisms of primary or secondary cytotoxicity, is critical in clinical research. How tumor cells obtain chemoresistance in response to chemoagent treatment is important.

It was suggested that circulating miRNAs (miRNA-122, -155, -146a, and -125b) presented in exosomes regulates inflammation in mouse models of alcoholic liver disease, drug-induced liver injury, and Toll-like receptor ligand-induced inflammatory cell-mediated liver damage [12]. Our results on elevated levels of miRNAs in secreted exosomes upon chemoagent treatment could be used to evaluate the context of cancer cell damage by chemoagents [13]. Another study in the literature also suggested that the secreted exosomes from anti-cancer-drug-treated hepatoma cells were functional [16] and that they contained heat-shock proteins that efficiently induced HSP-specific NK cell responses.

In this study, the isolated EVs from native or chemoagent-treated cells were characterized as exosomes with sizes around 30 nm in diameter that express the CD63 marker. There are different extracellular vesicles (EVs), ranging in different sizes [10,38]. Exosomes are identified as cell-derived microvesicles of 30~50 nm in diameter. They are presented in extracellular fluid, plasma, or urine samples, and are taken up by target cells [6,7]. The presence of specific proteins, lipids, mRNAs [6], or miRNAs [8] make them novel regulators, and they are involved in several physiological or pathological processes [9,10,11]. Exosomes are generated from late endosomes through the mechanism of multivesicular body (MVB) membrane formation. This is driven by a protein complex known as the endosomal sorting complex required for transport (ESCRT). However, evidence suggests that the raft microdomains enriched with sphingomyelinases and resulting ceramides are important for exosome formation and secretion [38]. Microvesicles (MVs) are relatively larger in size (100–1000 nm) and are generated by the outward budding of phosphatidylserine/cholesterol-enriched components [39]. Apoptotic bodies are released from the plasma membrane of apoptotic cells, and they range in the size of 1~5 mm. These closed structures contain most of the cellular components including organelles [40]. Although cell apoptosis is the dominant phenomenon for cancer cells upon chemoagent treatment, our isolation procedure enriched exosomes but not apoptotic bodies. Nevertheless, we showed that the intracellular level of miR-1976 was reduced but the extracellular level of miR-1976 increased (Figure 3). This suggested that the redistribution of miR-1976 from inside to outside compartments of cancer cells is a general phenomenon associated with chemoagent-induced cell apoptosis. This is explained by the deceased levels of miR-1976 achieving sufficient levels of XAF1 to sustain cell apoptosis. However, the secreted exosomes might be taken up by the surrounding tumor cells, which would reduce sensitivity toward chemoagents. The detailed mechanisms and exact roles of secreted exosomes should be investigated in the future.

In this study, we showed the drug response of hepatoma and pancreatic cancer cells to increased p21 expression (Figure 4). However, the role of p21 in cancer is complex and context-dependent [41,42,43]. The p21 protein plays a critical role in regulating cell cycle progression and DNA repair. In normal cells, p21 is induced in response to DNA damage or other stresses, leading to cell cycle arrest and successive DNA repair. However, in cancer, the functions of p21 depend on the stage and type of cancer. It can be tumor-suppressive or oncogenic. The p21 protein acts as a tumor suppressor by inhibiting the activity of cyclin-dependent kinases (CDKs) that drive cell cycle progression. Loss or downregulation of p21 expression can lead to uncontrolled cell proliferation and tumor growth. On the other hand, in some types of cancer, p21 can be oncogenic, promoting tumor growth and survival, inhibiting cell apoptosis, activation of the PI3K/Akt/mTOR pathway, and induction of angiogenesis. Inhibiting p21 expression also contributes to chemoresistance by prompting repair. However, we observed increased p21 expression in response to chemoagent treatments (Figure 4). An increased level of p21 would be protective, which leads to cell cycle arrest and delayed progression (Appendix A). However, our results did not suggest a mechanism for AT-1976 in promoting the p21 effect (Appendix A).

In this paper, we demonstrated the release of exosomes from anti-cancer drug-treated cancer cells that contained enriched miRNAs, especially miR-1976. This specific miR-1976 would regulate expression of the proapoptotic XAF1 protein, and thus reduce the levels of cell apoptosis or cell death. There are few studies related to miR-1976, and its role as a tumor suppressor or oncogene is not clear. It was suggested that miR-1976 could function as a tumor suppressor that regulates PLCE1 in non-small cell lung cancer. Down-regulation of miR-1976 was associated with TNM stage and postoperative survival, while miR-1976 overexpression would inhibit tumor growth and metastasis of NSCLC cells [44]. However, another study that showed several plasma miRNAs, including miR-1976, were upregulated in patients with metastatic melanoma [45]. Based on our preliminary results, we suggest that miR-1976 may act as an oncogene in HCCs since its presence would desensitize the hepatoma cells from chemoagents, which would promote tumor growth by suppressing basal cell apoptosis.

We also proposed a novel approach for mining cell-specific miRNA targets with biological relevance and functional correlation (Figure 1B). This approach not only enriches microRNA-specific target pools, but also provides binding site information. An additional advantage of this approach is that it reveals targeting sites in situ, which is dependent on the affinity of miRNA–mRNA interaction, and the abundance of target mRNAs in one specific cell. For instance, the *XAF1* segment was targeted and cloned from HepG2 cells (*XAF1*-positive) but not from Hep3B cells (*XAF1*-negative), as presented in Appendix A. This correlates exactly with the content of *XAF1* mRNA and protein expression (Figure 2A). In addition, the targeted sites revealed by in situ mRNA target screening were consistent with bioinformatic prediction in silico. Our mining strategy would be useful for miRNA–target validation or screening, since this provides cell-specific miRNA targets with binding site information experimentally.

Finally, we also suggest the potential application of this study in cancer therapy by interfering with the miR-1976-mediated responses through the use of anti-miR-1976 nucleic acid, or so-called antagomir [35]. Antagomirs are synthetic RNAs complementary to specific miRNAs. Chemical modification in the form of 2′-methoxy groups at each ribose and phosphorothioate group at the 5′ and 3′ terminus of the RNA protect it from Ago2 degradation and extends its half-life. In addition, 3′-lipid modification also enhances its cellular uptake. Recently, many papers have demonstrated the therapeutic applications of antagomirs interfering with miRNA function. For instance, antagomiR-613 protected neuronal cells from oxygen glucose deprivation/re-oxygenation by increasing SphK2 expression [46]. The miR-328 antagomir improved erectile dysfunction in streptozotocin-induced diabetic rats by regulating cGMP and AGEs [47]. AntagomiR-451 inhibited oxygen glucose deprivation-induced HUVEC necrosis by activating AMPK signaling [48]. An antagomir against miR-106b-5p ameliorated cerebral ischemia and reperfusion injury in an animal model through the inhibition of apoptosis and oxidative stress [49]. Antagomir-1290 suppressed CD133(+) cells in non-small cell lung cancer by targeting the fyn-related Src family tyrosine kinase [50]. Administration of antagomir-223 could inhibit apoptosis and promote angiogenesis and functional recovery of rats with spinal cord injury [51].

In summary, we propose a novel mechanism by which the intracellular level of miR-1976 determines the cell’s chemosensitivity. There is the potential to apply miR1976-specific antagomirs in combination with chemoagents to treat *XAF1*^+^ gastrointestinal cancers such as pancreatic cancer or liver cancer. The possible strategy of blocking miR-1976 biosynthesis as well as elevating its distribution outside cancer cells could develop as a potential therapeutic application against neoplasia. The understanding of this mechanism will be beneficial for developing novel strategies in cancer treatment.

## Figures and Tables

**Figure 1 biomedicines-11-01136-f001:**
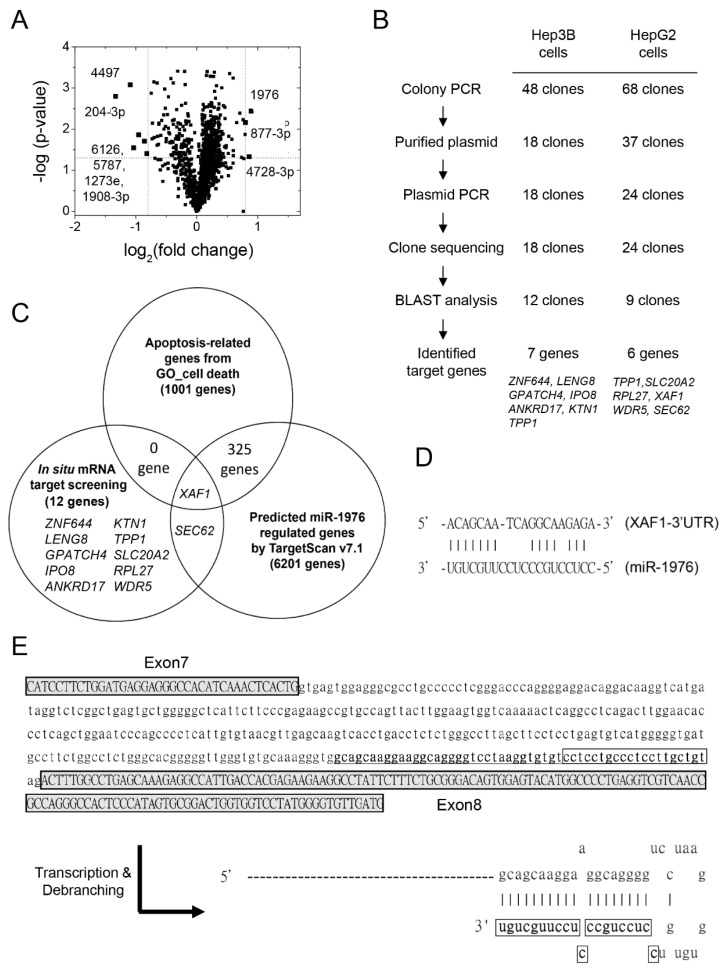
Increased levels of exosomal miRNA-1976 in chemoagent-treated hepatoma-derived exosomes might regulate proapoptotic XAF1 expression. (**A**) Volcano plot showing upregulation of hsa-miR-1976, -877-3p, and -4728-3p and downregulation of hsa-miR-4497, -204-3p, -6126, -5787, -1273e, and 1908-3p in chemoagent-treated hepatoma-derived exosomes. (**B**) Strategy and procedure for in situ mRNA target screening by miR-1976-specific cloning for HepG2 and Hep3B cells. (**C**) Intersection of the gene lists belonging to apoptosis-related genes, miR-1976 regulated genes, and target-screened genes in situ. The list suggests that XAF1 is a potential target regulated by miR-1976. (**D**) The mRNA–miRNA duplex shows an energy of –17 Kcal/mol as determined by DIANNA. (**E**) pre-miR-1976 (underlined) is located at the 3′-terminus of the intron segment between exon 7 and exon 8 of the RPS6KA1 gene. The boxed sequence is the mature miR-1976. Debranching of the RNA transcript generates a hairpin structure with a 5′-tail.

**Figure 2 biomedicines-11-01136-f002:**
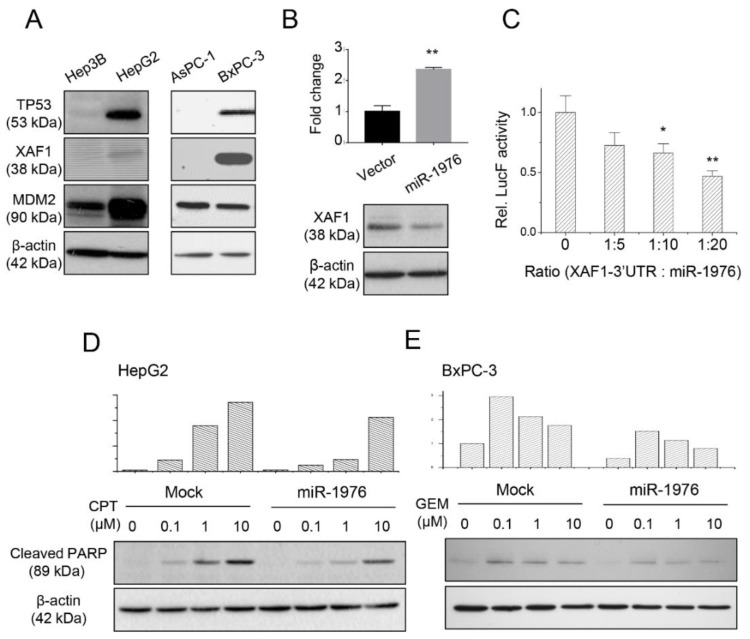
XAF1 is a potential target of miR-1976 as validated by Western blotting and a luciferase binding assay. (**A**) Expression of p53-dependent apoptotic proteins (TP53, XAF1, and MDM2) in hepatoma and pancreatic cancer cells. (**B**) Transfection of the miR-1976 expression plasmid increased the level of miR-1976 and decreased XAF1 expression. (**C**) Luciferase-binding assay showing dose-dependent inhibition of luciferase activity by miR-1976. Data are mean ± S.D. (*n* = 3); * *p* < 0.05; ** *p* < 0.01. The miR-1976 suppressed chemoagent-induced cell apoptosis in (**D**) hepatoma HepG2 and (**E**) pancreatic BxPC-3 cancer cells.

**Figure 3 biomedicines-11-01136-f003:**
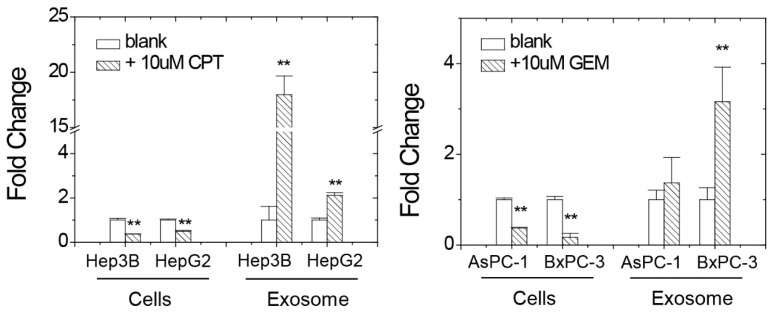
miR-1976 levels inside the cells and in the exosomes with or without CPT or GEM chemoagent treatment in hepatoma Hep3B/HepG2 cells and pancreatic AsPC-1/BxPC-3 cells as analyzed by qPCR. Data were mean ± S.D. (*n* = 3); ** *p* < 0.01.

**Figure 4 biomedicines-11-01136-f004:**
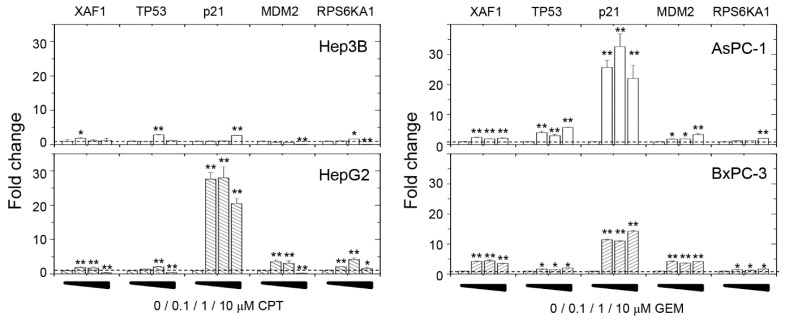
Effects of chemoagent treatment on gene expression levels in hepatoma Hep3B/HepG2 cells and pancreatic AsPC-1/BxPC-3 cells as analyzed by qPCR. Data are mean ± S.D. (*n* = 3); * *p* < 0.05; ** *p* < 0.01.

**Figure 5 biomedicines-11-01136-f005:**
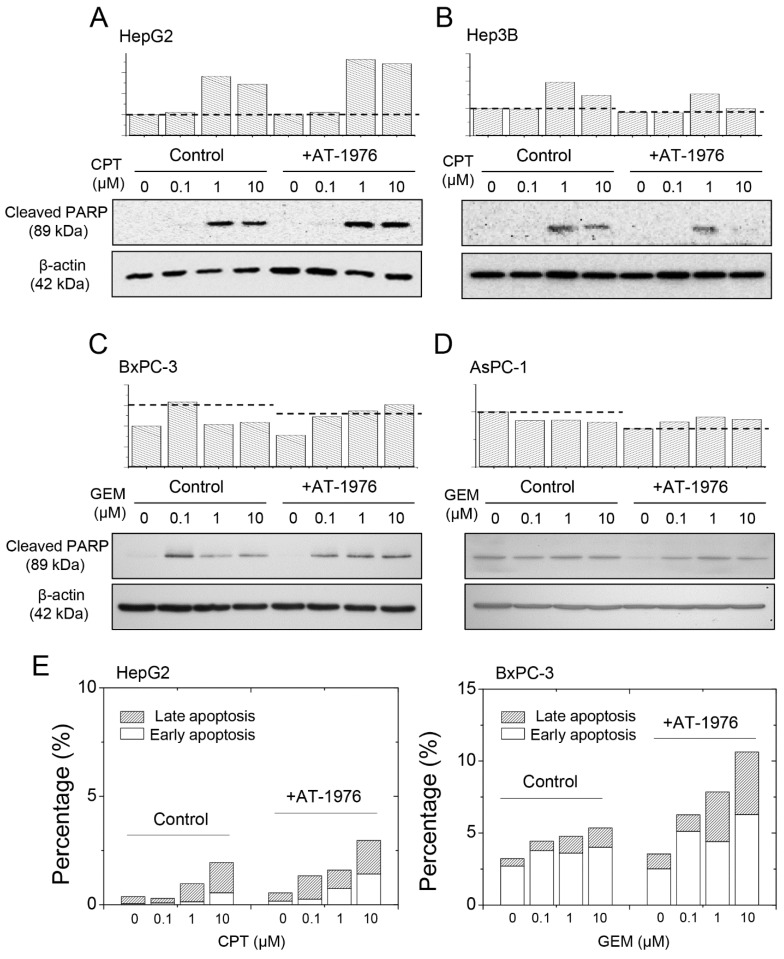
AT-1976 enhanced chemoagent-induced cell apoptosis in *XAF1*^+^-cells (HepG2 and BxPC-3 cells), but not in XAF1^−^ cells (Hep3B and AsPC-1 cells). The levels of cleaved PARP were examined by Western blot analysis in (**A**) HepG2, (**B**) Hep3B, (**C**) BxPC-3, and (**D**) AsPC-1 cells. (**E**) Cell apoptosis evaluated by annexin V and PI staining. Gradual increases in percentages of total apoptosis (early apoptosis + late apoptosis) as examined by flow cytometry.

**Figure 6 biomedicines-11-01136-f006:**
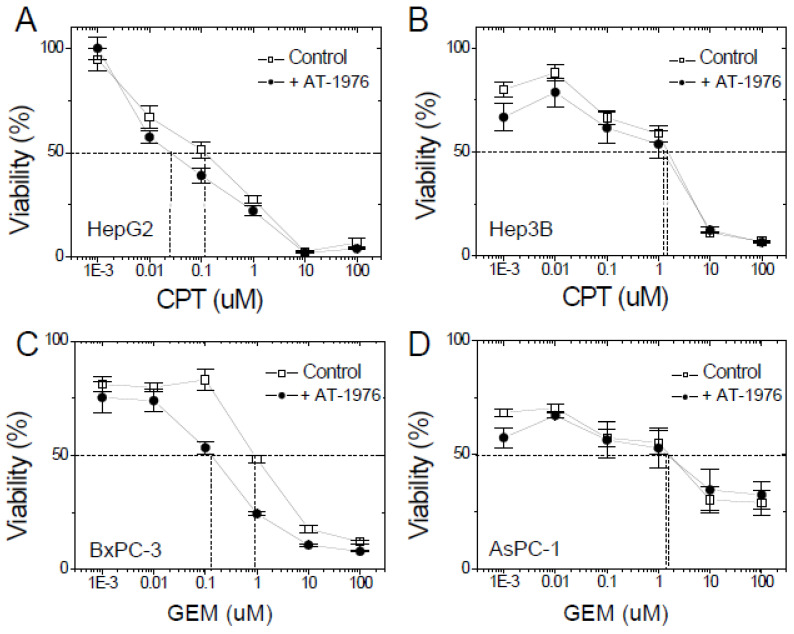
AT-1976 enhanced the chemosensitivity of *XAF1*^+^-cells (HepG2 and BxPC-3 cells) but not of XAF1^−^-cells (Hep3B and AsPC-1 cells), as evaluated by a cell viability assay. Cell viability was examined by the Alamar Blue assay in (**A**) HepG2, (**B**) Hep3B, (**C**) BxPC-3, and (**D**) AsPC-1 cells. Data were mean ± S.D. (*n* = 6).

**Figure 7 biomedicines-11-01136-f007:**
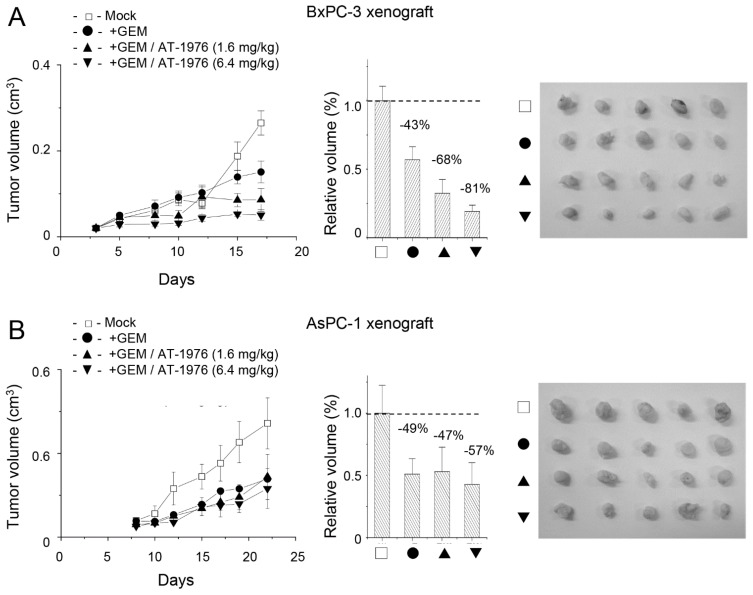
AT-1976 enhanced the efficacy of GEM in inhibiting tumor growth in a BxPC-3 xenograft (**A**), but not in a AsPC-1 xenograft (**B**). Each group contained 5 mice. Data were mean ± S.D. (*n* = 5).

## Data Availability

The array data were deposited in the NCBI Gene Expression Omnibus (GEO) database (http://www.ncbi.nlm.nih.gov/geo/ (accessed on 29 January 2023)) with the accession number GSE74829.

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
