# Peer review of "Anti-microRNA-1976 as a Novel Approach to Enhance Chemosensitivity in XAF1+ Pancreatic and Liver Cancer"

_biomedicines, 2023, doi:10.3390/biomedicines11041136_

Round 1
Reviewer 1 Report
The authors studied miRNA-1976 and its antagonist's effect on the chemo-sensitization of XAF1-positive gastrointestinal cancer. They found that miR-1976 was highly present in exosomes after chemo treatment and reduced inside cancer cells. They developed a new approach for in situ mRNA target screening and identified a few miR-1976-specific mRNA targets. The authors found that XAF1 is a significant target of miR-1976 and their interaction suppressed chemo agent-induced cell apoptosis in gastrointestinal cancer. They showed that the miR-1976 antagonist, AT-1976, enhanced chemo-sensitivity in hepatoma and pancreatic cancer cells in a XAF1-dependent manner. This was evidenced by increased cell apoptosis, reduced toxicity, and suppressed tumor growth in animal xenograft experiments in vivo. Overall, the study is interesting. However, the below points need to be clarified in the revised version.
1) Performing a cell apoptosis assay along with PARP cleavage analysis would strengthen the manuscript.
2) The manuscript should mention the statistical analysis and the number of times the western blots were repeated.
3) In Figure 4, the manuscript shows that p21 expression levels increased upon chemo agent treatment, but it did not clearly discuss p21's role in this context, which could be both pro-apoptotic or anti-apoptotic.
4) The manuscript should mention the number of mice in each group and the method of distribution used in the animal experiments.
5) Tumor-representative images should be presented along with the graphs.
Author Response
1) Performing a cell apoptosis assay along with PARP cleavage analysis would strengthen the manuscript.
Ans: Apoptotic analysis (Annexin and PI staining) was performed and the results were shown in Figure 5E and Supplementary Figure 4.
2) The manuscript should mention the statistical analysis and the number of times the western blots were repeated.
Ans: Descriptions on statistical analysis and repeated western blot were added at the method part (see line 171 and line 235).
3) In Figure 4, the manuscript shows that p21 expression levels increased upon chemo agent treatment, but it did not clearly discuss p21's role in this context, which could be both pro-apoptotic or anti-apoptotic.
Ans: We descript p21’s role, please check line518-534
4) The manuscript should mention the number of mice in each group and the method of distribution used in the animal experiments.
Ans: Thank you for reminding. We add description in the method section (page 5) and Figure legend of Fig.7. “Each group contained 6 mice. Data were mean ± S.D. (n=5).”
5) Tumor-representative images should be presented along with the graphs.
Ans: For your request, we added the images in Figure 7
Reviewer 2 Report
MicroRNAs that originate without Drosha cleavage and from the intron splicing are called mirtrons. The microRNA studied by the authors is a mirtron. miR-1976 is a 5p tailed mirtron from the gene RPS6KA1. I highly recommend the authors to mention that miR-1976 is a mirtron at appropriate places in the manuscript.
I request the authors to perform apoptotic functional TUNEL assay with and without anti-miR-1976 molecules and include this data in the manuscript to substantiate their findings.
I suggest the authors to include the chromosomal location of both miR-1976 and RPS6KA1 genes in a figure.
I request the authors to perform cell cycle analysis and apoptotic analysis (Annexin and Propidium Iodide) by flow cytometry with and without anti-miR-1976.
I suggest the authors to mention the molecular weight of the proteins probed as kDa in all the western blotting images.
Author Response
1) MicroRNAs that originate without Drosha cleavage and from the intron splicing are called mirtrons. The microRNA studied by the authors is a mirtron. miR-1976 is a 5p tailed mirtron from the gene RPS6KA1. I highly recommend the authors to mention that miR-1976 is a mirtron at appropriate places in the manuscript.
Ans: Thank you for this useful suggestion. We added a description at line 325-329.
2) I request the authors to perform apoptotic functional TUNEL assay with and without anti-miR-1976 molecules and include this data in the manuscript to substantiate their findings.
Ans: Thank you for the suggestion. We didn’t perform the TUNEL assay since cleaved PARP levels could represent the contents of late apoptosis.
3) I suggest the authors to include the chromosomal location of both miR-1976 and RPS6KA1 genes in a figure.
Ans: We had added a description and Figure 1E for your request.
4) I request the authors to perform cell cycle analysis and apoptotic analysis (Annexin and Propidium Iodide) by flow cytometry with and without anti-miR-1976.
Ans: Apoptotic analysis (Annexin and PI staining) was performed and the results were shown in Figure 5E and Supplementary Figure 2.
5) I suggest the authors to mention the molecular weight of the proteins probed as kDa in all the western blotting images.
Ans: Thank you for the suggestion. We add the information at the method part. Please see line 169-171.
Reviewer 3 Report
Manuscript entitled "Anti-microRNA-1976 as a novel approach to enhance chemosensitivity in XAF1+ gastrointestinal cancer"
Major issues:
1. All four cell lines used are hepatobiliary tract cancer instead of "GI" cancer. The title should be modified.
2. The author should add validation in real clinical cohort to confirm the clinical significance of microRNA-1976 in cancer tissue that should in parallel with the cell lines (i.e. Hepatoma, hepatoblastoma, and pancreatic cancer).
Author Response
1. All four cell lines used are hepatobiliary tract cancer instead of "GI" cancer. The title should be modified.
Ans: Thank you for the suggestion. Pancreatic cancer is not generally considered to be hepatobiliary tract cancer. Although the pancreas is located near the liver and bile ducts, it is not part of the hepatobiliary tract. Accordingly, we revised our title into:
“Anti-microRNA-1976 as a novel approach to enhance chemosensitivity in xaf1+ pancreatic and liver cancer”
2. The author should add validation in real clinical cohort to confirm the clinical significance of microRNA-1976 in cancer tissue that should in parallel with the cell lines (i.e. Hepatoma, hepatoblastoma, and pancreatic cancer).
è Currently, we can’t afford a clinical cohort directly relating to our issue. We did explore some websites on microRNA expression at cell lines. The miR-1976 expression presented in Cancer Cell Line Encyclopedia (CCLE; https://sites.broadinstitute.org/ccle/datasets) showed low expression of miR-1976 under normal, untreated conditions. However, we claimed that chemoagent treatments increased and released miR-1976 through exosomes. So that, we can’t provide further information now.
Reviewer 4 Report
It was showed that miR-1976 was highly enriched in secreted exosomes upon chemoagent treatments through microarray analysis. qPCR analysis confirmed the increased levels of miR-1976 in secreted vesicles and the reduced levels inside the cancer cells after the treatments. One novel approach of in situ mRNA target screening by gene-specific cloning and sequencing was developed. Several miR-1976-specific mRNA targets were revealed. Of them, the proapoptotic gene target, XAF1, was confirmed, and this interaction suppressed chemoagent-induced cell apoptosis. The response to chemoagent-induced cell stress and increased production of RPS6KA1 gene transcription was associated with the increase in its intronic pre-miR-1976 expression. Secretion of miR-1976 outside cancer would be a general phenomenon responsive to chemoagent treatments. In addition, we further demonstrated the application of miR-1976 antagonist, AT-1976, could enhance chemosensitivies of hepatoma and pancreatic cancer cells in a XAF1-dependent manner, as evidenced by increased levels of cell apoptosis, reduced IC50 in cell toxicity assays, and suppressed tumor growth in animal xenograft experiments in vivo. The manuscript is interesting and the research was well performed. A novel mechanism that intracellular level of miR-1976 determined the chemosensitivity, and propose a strategy that blockade of miR-1976 biosynthesis as well as the elevated distribution outside cancer cells, could be develop as one potential therapeutic application against neoplasia.
At the end of introduction part there is need to focuses more on the novelty of present research. It should not contain the conclusions of the performed research.
Overall conclusion at the end of manuscript would be desirable.
Author Response
1) At the end of introduction part there is need to focuses more on the novelty of present research. It should not contain the conclusions of the performed research.
Ans: We add description at final part of discussion. Please see line 51-53.
2) Overall conclusion at the end of manuscript would be desirable.
Ans: We add description at final part of discussion. Please see line 577-583.
Round 2
Reviewer 2 Report
I suggest the authors to add the molecular weight of the proteins in kDa in all the western blot images given as figures. I see that the authors have added the same in the methods section which will be difficult for the readers to know when they look at the figures.
Author Response
Thank you for the suggestion. We add the molecular weights of the blots in Figure 2 and 5.
Reviewer 3 Report
The revision might reach the standard to be accepted.
Author Response
Thank you very much!